# Mechanisms of Cell Death Induced by Cannabidiol Against Tumor Cells: A Review of Preclinical Studies

**DOI:** 10.3390/plants14040585

**Published:** 2025-02-14

**Authors:** Edilene S. A. Melo, Estefani A. Asevedo, Joaquim Maurício Duarte-Almeida, Fahrul Nurkolis, Rony Abdi Syahputra, Moon Nyeo Park, Bonglee Kim, Renê Oliveira do Couto, Rosy Iara Maciel de A. Ribeiro

**Affiliations:** 1Experimental Pathology Laboratory, Dona Lindu Central-West Campus (CCO), Federal University of São João del-Rei (UFSJ), Sebastião Gonçalves Coelho 400, Chanadour, Divinopolis 35501-296, MG, Brazil; melo.edilene97@gmail.com (E.S.A.M.); asevedoestefani@gmail.com (E.A.A.); 2Plant Cell Culture Laboratory, Dona Lindu Central-West Campus (CCO), Federal University of São João del-Rei, Sebastião Gonçalves Coelho 400, Chanadour, Divinopolis 35501-296, MG, Brazil; maudall@ufsj.edu.br; 3Department of Biological Sciences, Faculty of Sciences and Technology, State Islamic University of Sunan Kalijaga (UIN Sunan Kalijaga), Yogyakarta 55281, Indonesia; fahrul.nurkolis.mail@gmail.com; 4Department of Pharmacology, Faculty of Pharmacy, Universitas Sumatera Utara, Medan 20155, Indonesia; rony@usu.ac.id; 5Department of Pathology, College of Korean Medicine, Kyung Hee University, Seoul 02447, Republic of Korea; mnpark@khu.ac.kr (M.N.P.); bongleekim@khu.ac.kr (B.K.); 6Pharmaceutical Development Laboratory, Dona Lindu Central-West Campus (CCO), Federal University of São João del-Rei, Sebastião Gonçalves Coelho 400, Chanadour, Divinopolis 35501-296, MG, Brazil; rocouto@ufsj.edu.br

**Keywords:** cannabidiol, antineoplastic agents, transient receptor potential channels, preclinical drug evaluation

## Abstract

Commonly known as marijuana or hemp, *Cannabis sativa* L. (Cannabaceae), contains numerous active compounds, particularly cannabinoids, which have been extensively studied for their biological activities. Among these, cannabidiol (CBD) stands out for its therapeutic potential, especially given its non-psychotropic effects. This review evaluates the antitumor properties of CBD, highlighting its various mechanisms of action, including the induction of apoptosis, autophagy, and necrosis. By synthesizing findings from in vitro studies on the cell death mechanisms and signaling pathways activated by CBD in various human tumor cell lines, this literature review emphasizes the therapeutic promise of this natural antineoplastic agent. We conducted a comprehensive search of articles in PubMed, Scopus, Springer, Medline, Lilacs, and Scielo databases from 1984 to February 2022. Of the forty-three articles included, the majority (68.18%) reported that CBD activates apoptosis, while 18.18% observed simultaneous apoptosis and autophagy, 9.09% focused on autophagy alone, and 4.54% indicated necrosis. The antitumor effects of CBD appear to be mediated by transient receptor potential cation channels (TRPVs) in endometrial cancer, glioma, bladder cancer, and myeloma, with TRPV1, TRPV2, and TRPV4 playing key roles in activating apoptosis. This knowledge paves the way for innovative therapeutic strategies that may enhance cancer treatment outcomes while minimizing the toxicity and side effects associated with conventional therapies.

## 1. Introduction

Cannabidiol (CBD) is one of the main cannabinoids found in marijuana or hemp (*Cannabis sativa* L.), a plant with both medicinal and psychoactive properties. Its use as a medicine dates back to 500 BC in Asia, and it has been extensively studied since the 1960s [1,2]. Approximately 554 chemical compounds have been identified in this species. Of the one hundred thirteen phytocannabinoids, two stand out: CBD and tetrahydrocannabidiol (THC). These cannabinoids exhibit both similar and distinct bioactive effects. CBD is known as an antitumor agent and lacks psychoactive effects, making it more suitable for clinical applications compared to Δ9-tetrahydrocannabinol (Δ9-THC), which is known for its psychotropic effects. Thus, compared with THC, CBD is more advantageous for clinical applications due to its antitumor properties [3,4]. Despite the remarkably similar chemical structures (C21H30O2) of these two cannabinoids, there is a key difference. THC contains a cyclic ring, while CBD has a hydroxyl group. The psychoactive and antitumor effects of THC are due to its interaction with specific cannabinoid type 1 (CB1) receptors, which are expressed throughout the central nervous system. In contrast, its immunomodulatory effects are mediated through cannabinoid type 2 (CB2) receptors, which are found in the immune system and hematopoietic cells [5]. However, the intoxicating properties of Δ9-THC limit its therapeutic use as an isolated agent.

The molecular mechanisms by which CBD exerts its antitumor effects mostly seem to be independent of the CB1 and CB2 receptors. Other receptors implicated in CBD action include transient receptor potential cation channels (TRPVs), such as TRPV1 and TRPV2, GPR55, VDAC1, CNR1/CB1, CNR2/CB2, SLC8A1, TRPC, and TRPM [6]. Some studies suggest that CBD acts as an agonist of TRPVs, which may be involved in tumor cell proliferation, apoptosis, angiogenesis, migration, and invasion [6,7]. Among the hallmarks of cancer is the ability to sustain proliferative signaling and resist cell death [8]. Therefore, many therapies are designed to cause damage that leads to cell death [9]. Research efforts have focused on understanding how CBD can promote cell death, which may occur through apoptosis, autophagy, a combination of both, or necrosis [10].

Apoptosis, or programmed cellular self-destruction, involves the synthesis of proteins, complex apoptotic machinery, and numerous signaling pathways [11,12]. Apoptotic signaling pathways are typically activated by cleaved caspases, which are endoproteases [13]. Caspases are classified based on their positions in the apoptotic signaling cascade, as initiators (caspase-1, caspase-2, caspase-4, caspase-5, caspase-8, caspase-9, caspase-10, caspase-11, and caspase-12) or effectors (caspase-3, caspase-6, and caspase-7). Once cleaved, these caspases activate each other, resulting in the amplification of their activities through a protease cascade [14]. Caspase activation can be initiated by different pathways: via death receptors in the plasma membrane (the extrinsic pathway), in the mitochondria (the mitochondrial or intrinsic pathway), or through the endoplasmic reticulum (ER), which is part of the intrinsic pathway [15,16]. The intrinsic and extrinsic pathways may be interconnected and can be activated simultaneously [17].

Autophagy is a process by which cellular components such as macroproteins or organelles are sequestered by lysosomes for degradation in response to stress, nutrient deprivation, or damaged organelles. Lysosomes digest these components, which can be recycled to produce new cellular structures or organelles or even serve as a source of energy. Although cells generally use autophagy to recycle components, this process can also lead to cell death to eliminate aging tissues, senescent cells, or neoplastic lesions [18]. This self-degradation process activates a series of proteins, including AMPK, which inhibits the formation of the mTOR complex, thereby reducing the inhibitory effect of this complex on ULK1. This, in turn, promotes the production of autophagic vesicles through the activation of proteins such as Beclin and LC3, which are responsible for the elongation of autophagic vesicles and the formation of autophagosomes [19].

Apoptosis and autophagy can be activated simultaneously, sometimes working synergistically or opposing each other. Their concurrent activation can result in either cell death or survival, depending on whether autophagy acts as an antagonist to cell death [20].

Despite increasing evidence regarding the effects of CBD on tumor cells, the underlying mechanisms of its antitumor activity remain unclear. Mutation in specific genes that mediate death signaling can compromise treatment efficacy, such as chemotherapy [21]. Therefore, understanding the mechanism of cell death signaling pathways activated by CBD is relevant to designing possible targeted cancer treatments further. In this review, we gather and summarize evidence from preclinical in vitro studies reporting the cell death mechanisms activated upon treatment with CBD. Moreover, the receptor and molecular targets involved in the antitumor efficacy of CBD are presented.

## 2. Results

### 2.1. Molecular Docking of Cannabidiol and Receptors

The molecular docking analysis investigated the interactions between CBD and vanilloid receptors (Table 1) and CBD with cannabinoid and VDAC1 receptors (Table 2). Regarding vanilloid receptors, cannabidiol exhibited very similar docking scores. The interaction with TRPV1 (PDB ID: 8GF8) exhibited a high binding affinity, −7.9 kcal/mol, and CBD formed contact with amino acids in Chain A (PHE43, ASN438, and VAL441) and Chain B (PHE587 and PHE591). The interaction with TRPV4 (PDB ID: 8T1B) demonstrated strong binding affinity, as indicated by a docking score of −7.8 kcal/mol, and involved interactions with amino acids in Chain A, namely TYR439, PHE471, and ASN474. Among the vanilloid receptors, TRPV2 (PDB ID: 2F37) exhibited the lowest docking score with CDB (−7.5 kcal/mol); however, this value still indicates a high affinity. Notably, TRPV2 is implicated in inducing the death of various cancer cell types; however, many studies also indicate that the action of CBD is also mediated by the receptors TRPV1, TRPV4, CB1, CB2, and VDAC1 (Figure 1).plants-14-00585-t001_Table 1Table 1Molecular docking of the interactions between cannabidiol with vanilloid receptors using CB-Dock2. https://cadd.labshare.cn/cb-dock2/index.php/ (accessed on 2 December 2024). Color coding in Cb-Dock2 images: Gray dashed lines, hydrophobic interactions or general molecular; Green dashed lines, hydrogen bonds; Blue dashed lines, pi-stacking (aromatic interactions) or electrostatic interactions; Gray structure, ligand; Surrounding structure, amino acid residues of the protein; red parts, oxygen atoms; blue parts, nitrogen atoms.Vanilloid ReceptorsVina Score (kcal/mol)Amino Acid InteractionVisualizationTRPV1−7.9Chain A: PHE435 ASN438 VAL441 TYR442 LEU444 TYR445 ILE448 LEU480 SER481 LEU483 GLY484 TYR487 PHE488 ARG491 TYR511 SER512 GLU513 LEU515 PHE516 LEU518 PHE522 PHE543 ALA546 LEU547 THR550 ASN551 LEU553 TYR554 TYR555 ARG557 ALA566 VAL567 ILE569 GLU570 ILE573 LEU574 GLN701 THR705 ASP708 THR709 SER712 PHE713 LYS715Chain B: PHE587 PHE591
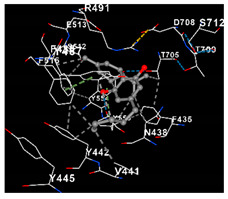
TRPV2−7.5Chain A: PRO71 ASN72 ARG73 PRO95 GLU96 TYR97 SER99 LYS100 THR101 ASP144Chain B: GLN239 THR283 VAL284 GLN285 LEU286 GLU287 ASP288 ILE289 ARG290 ASN291 LEU292 ASP294
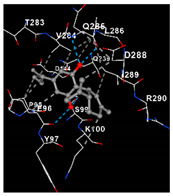
TRPV4−7.8Chain A: TYR439 PHE471 ASN474 SER477 TYR478 ALA481 PHE519 THR520 LEU523 PHE524 THR527 ASN528 ASP531 LYS535 LYS536 PRO538 ILE545 ASP546 PHE549 GLN550 TYR553 TYR591 PHE592 THR739 THR740 ASP743 ILE744 SER747
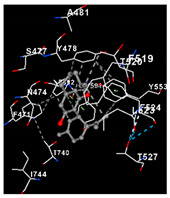

plants-14-00585-t002_Table 2Table 2Molecular docking of the interactions between cannabidiol with cannabinoid and VDAC1 receptors using CB-Dock2. https://cadd.labshare.cn/cb-dock2/index.php/ (accessed on 2 December 2024). Color coding in Cb-Dock2 images: Gray dashed lines, hydrophobic interactions or general molecular; Green dashed lines, hydrogen bonds; Blue dashed lines, pi-stacking (aromatic interactions) or electrostatic interactions; Gray structure, ligand; Surrounding structure, amino acid residues of the protein; red parts, oxygen atoms; blue parts, nitrogen atoms.Cannabinoid ReceptorsVina Score (kca/mol)Amino Acid InteractionVisualizationCB1−7.4Chain R: MET109 VAL110 LEU111 ASN112 PRO113 PHE170 SER173 PHE174 PHE177 PHE189 LYS192 LEU193 VAL196 ASP266 ILE267 PHE268 LYS376 PHE379 ALA380 SER383
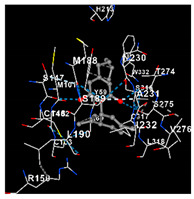
CB2−9.0Chain R: TYR25 MET26 PHE87 SER90 PHE91 PHE94 HIS95 PHE106 LEU107 LYS109 ILE110 GLY111 VAL113 THR114 PHE117 SER165 PRO168 GLU181 LEU182 PHE183 PRO184 ILE186 TYR190 LEU191 TRP194 TRP258 VAL261 LEU262 MET265 LYS278 LYS279 PHE281 ALA282 SER285 CYS288
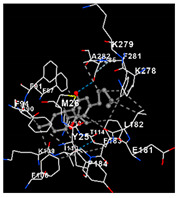
VDAC1−7.6Chain A: VAL206 ASN207 LEU208 ALA209 TRP210 ASN214 SER215 THR217 ARG218 PHE219 GLY220 ILE221 VAL237Chain B: LEU180 THR182 VAL184 PHE190 GLY191 GLY192 SER193 ILE194 VAL206 ASN207 LEU208 TRP210 ASN214 SER215 THR217 ARG218 PHE219 ASN239
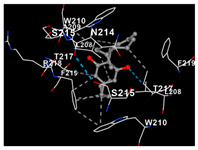


Regarding cannabinoid receptors, CBD demonstrated the highest binding affinity with CB2 (PDB ID: 8GUQ), with a docking score of −9.0 kcal/mol. It interacted exclusively with amino acids in Chain R, including TYR25, MET26, and PHE87. CBD also interacted with amino acids in Chain R of the CB1 receptor (PDB ID: 8IKG), such as MET109, VAL110, and LEU111. However, it showed lower affinity, as indicated by the docking score of −7.4 kcal/mol. VDAC1 receptors showed a very similar docking score of −7.6 kcal/mol, interacting with amino acids in Chain A, such as VAL206, ASN207, and LEU208, and in Chain B, such as LEU180, THR182, and VAL184 (Table 2).

### 2.2. The Role of Vanilloid Receptors in Cancer

Vanilloid receptors (TRPV) belong to one of the six subfamilies of TRP (transient receptor potential) cation channels. These ion channels are involved in various physiological processes, including nociception, thermosensitization, and renal Ca^2+^ uptake/reabsorption. TRPV1 and TRPV2 are cation channels activated by the endocannabinoids 2-AG and anandamide, as well as by Δ9-THC and CBD. Both TRPV1 and TRPV2 are widely distributed in the nervous system [22].

TRPV1 is expressed not only in the nervous system but also in other tissues, such as the epithelium, and it has a 40% sequence similarity with TRPV3. TRPV4, on the other hand, is a polymodal channel with a wide range of activation mechanisms. It is expressed in various tissues, particularly muscle and epithelial tissues [23].

The TRPV1, TRPV2, and TRPV4 receptors play a significant role in cancer pathology. While these receptors are well studied for their functions in sensory physiology, they are increasingly associated with cancer progression and development. Evidence suggests that TRPV1, TRPV2, and TRPV4 are overexpressed in several tumor types, contributing to processes such as tumor survival, proliferation, metastasis (migration and invasion), and angiogenesis [24,25,26].

Interestingly, both antagonists and agonists of TRPV receptors, including CBD, have demonstrated significant antitumor activity. CBD, acting as an agonist of TRPVs, has been shown to inhibit cell proliferation and activate cell death pathways, highlighting its potential role in cancer therapy [27].

Few studies have reported the detailed mechanisms of interaction between CBD and TRPV receptors. However, it is known that the activation of TRPV1 is associated with the production of reactive oxygen species (ROS) due to Ca^2+^ imbalance, which impairs mitochondrial membrane potential. Additionally, the activation of TRPV4 facilitates Ca^2+^ entry into cells, potentially leading to endoplasmic reticulum stress [28]. This evidence may justify the cytotoxic effects of CBD on tumor cells.

Pumroy and collaborators used cryoelectron microscopy to demonstrate that CBD interacts with TRPV2 through a hydrophobic pocket comprising hydrophobic and aromatic residues located in the S5 and S6 helices, distinct from the binding sites of other ligands and lipids. Electrophysiological experiments further revealed that CBD (10 μM) acts synergistically with 2-aminoethoxydiphenyl borate (2-APB), increasing the current density of the TRPV2 receptor by tenfold, thus serving as a potent agonist. Moreover, in silico docking studies have highlighted the importance of histidine521 and arginine539 residues in the activation of TRPV2 by CBD in combination with 2-APB. These findings underscore the need for further studies to fully elucidate the pharmacology of CBD as a ligand for TRPVs, given their potential as critical targets for future CBD-based treatments [29,30].

### 2.3. Cannabidiol Induces Death in Cancer Cells In Vitro and In Vivo

CBD has been shown to induce tumor cell death through apoptosis, autophagy, necrosis, and simultaneous activation of apoptosis and autophagy. Among the selected studies, summarized 30 studies demonstrating that CBD activates apoptosis in various tumor types. Four studies reported that CBD induces autophagy, two studies indicated that CBD triggers necrosis, and eight studies showed the simultaneous activation of apoptosis and autophagy by CBD.

Figure 2 and Figure 3 illustrate the underlying mechanisms associated with apoptotic and autophagic cell death, respectively. Additionally, Figure 4 provides an overview of the mechanisms involved in cell death mediated by both apoptosis and autophagy.

There is compelling evidence that the antitumor effects of CBD are mediated by its interaction with TRPVs in endometrial cancer, glioma, bladder cancer, and myeloma. The most consistent findings include the activation of apoptosis and the interaction of CBD with TRPV1, TRPV2, and TRPV4.

#### 2.3.1. Apoptosis

Apoptosis, a programmed cell death, is often inhibited in cancer due to the overexpression of anti-apoptotic proteins and downregulation of pro-apoptotic proteins. However, most of the studies included here reported that CBD can effectively inhibit tumor cell proliferation by inducing apoptosis (Table 3) [1,31].

Several studies have demonstrated that CBD induced apoptosis in breast cancer cells. CBD (3 and 5 μM) upregulated PPARγ and downregulated mTOR and cyclin D1 in MDA-MB-231 and T-47D cells, respectively [32]. In MCF-7 cells, CBD (38.42–64.6 μM) not only promoted apoptosis, with the total number of apoptotic cells increasing from 5.94% to 15.38–58.94% (*p* < 0.0001), but it was also synergistic with chemotherapeutics, significantly increasing the number of apoptotic cells (*p* < 0.0001) [33].

Moreover, in MDA-MB-231 cells, CBD (1 μM) encapsulated in extracellular vesicles increased the population of cells in the G1 phase (*p* < 0.01), indicating apoptosis. When combined with doxorubicin (DOX) (500 nM), the treatment decreased Bcl-2 and mTOR expression and increased Bax and caspase-9 expression. When tested in in vivo studies, the combination of CBD (5 mg/kg) and DOX reduced tumor volume (*p* < 0.001, n = 4), increased the expression of BAX and cleaved caspase-3, and decreased Bcl2 and mTOR [34].

Furthermore, the inhibitory effects of CBD have been extensively studied in Gioma and glioblastoma models. The treatment with CBD (20 µM) combined with gamma irradiation (5 Gγ) led to the upregulation of active JNK1/2 and MAPKP38, particularly in U87MG cells, increasing apoptotic levels to 50% at 48 h and 90% at 72 h. [35]. The same authors later demonstrated that in other glioma cell lines (U118MG and T98G), treatment with CBD (10 µM), isolated or combined with irradiation (10 Gγ), and the ATM kinase inhibitor KU60019 (1–2 µM) induced apoptosis through the upregulation of TRAIL/TRAIL-R2 and promoted DR5 activation [36].

Massi et al. (2004) and Massi et al. (2006) reported that CBD promoted apoptosis in glioblastoma cell lines (U87MG and U373), with IC_50_ values of 26.2 ± 2.8 µM and 24.1 ± 2.16 µM, respectively. In U87MG cells, the treatment (25 µM) increased ROS generation (more than 200%, 5 h of CBD exposure); it also decreased glutathione levels significantly to 55% (*p* < 0.001) and activated key mediators of the extrinsic and intrinsic apoptotic pathways, such as caspase-8 (2.5–3.5-fold relative to untreated), caspase-9, and caspase-3. There was an increase in apoptotic cells by 51.58 ± 4.82%, and the activation of cell death was not reversed after the use of CB1 and CB2 receptor antagonists. CBD did not show cytotoxicity and did not alter ROS and glutathione levels when tested in the primary culture of glial cells. In addition, CBD at a concentration of 0.5 mg/mouse reduced tumor growth of U87 glioma cells by approximately 70% on day 18 of treatment (572 ± 147 mm^3^, n = 7) compared to untreated animals (1765 ± 259 mm^3^, n = 7). On day 23 of treatment, there was a smaller regression of 50% of the tumor (1210 ± 210 mm^3^, n = 7) compared to the control (2212 ± 256 mm^3^, n = 7) [37,38]. Similarly, CBD with an IC_50_ value of 0.6 uM (0.5–1.0) induced apoptosis, causing oxidative stress in U251 cells and elevating ROS levels by up to 40%. While cell viability at doses of THC (1.7 µM) and CBD (0.4 µM) separately was 71 ± 4% and 83 ± 5%, the combination of these doses potentiated their effects synergistically, reducing cell viability to 7 ± 4%. The reduction in cell viability was due to the activation of cell death by apoptosis; upregulated caspase-3, caspase-7, and caspase-9; and increased PARP levels [39].

CBD presented different IC_50s_ against human glioma cell lines: J3TBG (5.77 µg/mL ± 0.38), U87MG: (8.2 µg/mL ± 0.69), and U373MG: (4.94 µg/mL ± 0.79). At these concentrations, the treatment induced apoptosis via RIPK3 since, when a RIPK3 inhibitor (GSK872 1 µM) was used, CBD at its cytotoxic concentrations did not alter cell viability. VDAC1 was also involved in the CBD death mechanism because the use of an inhibitor of this receptor (DIDS 50 µM) together with CBD treatment restored cell viability. This suggests that CBD alters VDAC1 permeability, causing Ca^2+^ influx and mitochondrial dysfunction [31].

In a neuroblastoma cell line (SK-N-SH), CBD (7.5 and 10 μg/mL) caused morphological changes related to apoptosis, thus increasing the levels of late apoptotic cells from 35% to 63% and increasing cleaved caspase-3 expression. The treatment with 20 mg/kg of CBD for 14 days reduced the tumor in mice with a xenograft of SK-N-SH neuroblastoma cells (an untreated group with 4.28 cm^3^ and CBD group 2.31 cm^3^, n = 12 and *p* < 0.05) and showed the activation of cleaved caspase-3 (*p* < 0.001) [7]. In primary glioma stem cells (GSCs, 3832 and 387), the IC_50_ values were 3.5 μM (3.4–3.6) and 2.6 μM (2.5–2.7), respectively. Treatment with CBD (2 µM) inhibited the self-renewal of these cell lines in a ROS-dependent manner because the effects of CBD on the reduction in tumorspheres were partially reversed when an antioxidant, vitamin E (40 µM), was used, reducing the expression of Sox2, Id1, and p-STAT3 while increasing p38 MAPK levels. Intracranial GSC glioblastoma xenograft models were treated with CBD (15 mg/kg) and presented a significant improvement in survival, with the inhibition of p-AKT and Ki67 and activation of caspase-3. Regarding tumor growth, it was inhibited until the 22nd day of treatment; after that, the tumor presented an adaptation, and the growth rate increased, possibly due to drug resistance between the 22nd and 29th days [40].

In cells derived from glioblastoma multiforme (GBM) with similar properties to glioma-initiating cells, treatment with THC (0.83 µM), CBD (4.17 µM), and TMZ (100 µM) induced apoptosis, as indicated by the activation of caspase-3 and cleaved PARP. Nude mice with a xenograft of U87MG glioblastoma cells, when treated with the combination of 5 mg/mL of THC and CBD (1:1 and 1:4) with TMZ, presented a notable reduction in tumor volume and increased animal survival (*p* < 0.001, n = 6–7) [41]. Additionally, the combination of CBD (10 µM) and carmustine (200 µM) successfully overcame glioma stem cell (GSC) resistance to carmustine and induced apoptosis by increasing the dissipation of the mitochondrial membrane potential three-fold [10].

The effects of CBD were also investigated against leukemia, lung, endometrial, cervical, head and neck, colorectal, bladder, pancreatic, gastric, and prostate cancers.

Lung cancer cell lines (A549 and H460), treated with CBD IC_50_ (3.47 and 2.80 μM), respectively, activated COX-2- and PPARγ-dependent apoptosis because the use of COX-2 (NS-398) and PPAR-γ (GW9662) inhibitors removed the cytotoxic and apoptotic effect of CBD. COX-2 activation by CBD was confirmed by the upregulation of prostaglandins PGD 2 (0.52 ± 0.20 ηmol/L in A549 and 0.14 ± 0.01 ηmol/L in H460) and 15d-PGJ 2 (0.94 ± 0.12 ηmol/L in A549 and 0.23 ± 0.05 ηmol/L in H460, n = 4. The action of CBD on primary lung cancer cells was also evaluated. There was a reduction in cell viability to 36.4% ± 3.2% compared to the 100% control with an IC_50_ of 0.124 µM in a manner dependent on COX-2 and PPAR-γ. Furthermore, in nude mice with A549 xenografts, CBD (5 mg/Kg) treatment led to the upregulation of cyclooxygenase-2 (COX-2) and PPARγ (an approximate increase of 50%, n = 5) and reduced tumor volume (*p* < 0.05, n = 5–7) [42]. In lung cancer stem cells and adherent lung cancer cells (A549, H1299, and SCLC H69), CBD (10 µM) activated effector caspases (caspase-3 and caspase-7), increased the expression of pro-apoptotic proteins, elevated ROS levels by 2.5-fold in A549 and H1299 cells (*p* < 0.0001 and *p* < 0.05), and caused a significant 60% loss of mitochondrial membrane potential in all non-adherent cell lines when compared to untreated cells (*p* < 0.001 and *p* < 0.0001) [43].

TRPV1 was upregulated in endometrial cancer cell lines Ishikawa treated with CBD (5 µM). It also increased ROS production (*p* < 0,05), chromatin condensation, and a 24% decreased mitochondrial membrane potential. The apoptotic cell death in the Ishikawa cell line was evidenced by elevated levels of caspase-3 (increased 39%), caspase-7 (increased 23%), and cleaved PARP. CBD was selective for tumor cells and did not exert cytotoxicity on the human foreskin fibroblast cell line (HFF-1) [44].

In cervical cancer cells, CBD exerted cytotoxicity against HeLa, SiHa ME-180 cell lines with IC50 values of 3 µg/mL, 2 µg/mL, and 1.5 µg/mL. The treatments induced apoptosis through the overexpression of p53, caspase-3, caspase-7, caspase-9, and Bax [45].

In a head and neck squamous cell carcinoma (HNSCC) cell line (SCC25), treatment with CBD (30 µM, *p* < 0.001 and n = 5) activated apoptosis, as demonstrated by annexin V staining. There was a significant increase in cells in the late apoptotic stage by 78.6% after treatment with CBD [46]. In endothelial cells associated with sarcoma (herpes virus positive), CBD exerted cytotoxicity on 50% of the cells at a dose of 2.08 uM, and CBD (2.0 and 10 µM) preferentially induced apoptosis with an increase of 80% to 100% of apoptotic cells. CBD cytotoxicity was preferential to Kaposi’s sarcoma tumor cells when compared to normal endothelium [47].

In colorectal tumor cell lines (HCT116, DLD-1), CBD (6 µM) induced apoptosis through the intrinsic apoptotic pathway in a ROS- (increased by 300%) and Noxa-dependent manner, mediated via the endoplasmic reticulum. The overexpression of cleaved PARP and caspase 3, 8, and 9 confirmed apoptosis. BALB/c nude mice with HCT116 cell tumors treated with CBD (20 mg/kg) showed a significant reduction in tumor size (*p* < 0.05, n = 5) as a result of Noxa-dependent apoptosis activation [48].

In T24 bladder tumor cells, CBD (30 µM) triggered apoptosis, as indicated by annexin V staining +/PI—(11.4%, n = 3), through the influx of Ca^2+^ via TRPV2 channels. The involvement of the TRPV2 receptor in the action of CBD was confirmed when it was silenced, thus blocking the effect of CBD in inducing apoptosis by 89% (annexin-V-FITC+/PI-, 1.31% vs. 11.4% TRPV2 non-silenced group) [49]. Similarly, Chen et al. (2021) reported that CBD exerted cytotoxicity on bladder tumor cells with IC_50_ values of (T24, 10.85 ± 2.18 µM), UM-UC-3 (21.83 ± 1.32 µM), and 5637 (22.92 ± 0.97 µM). For the normal bladder cell line (SV-HUC-1), CBD exerted less cytotoxicity than in tumor lines, with an IC50 value of 40.68 ± 1.87 µM. The concentration of CBD (12 µM) inhibited T24 cell proliferation and induced apoptosis in 49.91% of cells in early and late death through inactivation of the PI3K/Akt/mTOR pathway. The treatment also reduced Bcl-2 expression by 0.51-fold and upregulated Bax, cytochrome c, and caspase-7 by 1.47-, 2.27-, and 1.01-fold [50].

CB2 receptors mediated apoptosis in leukemia cell lines (EL-4, Jurkat, and MOLT-4) treated with CBD (5 µM), and the level of apoptotic cells increased from 6.4% (untreated group) to 48.7% after 24 h of treatment. It also increased ROS generation (*p* < 0.05) and promoted the activation of cytosolic cytochrome *c*, caspase-8, caspase-9, and caspase-3. It was also reported in this study that CBD (12.5 and 25 mg/kg) reduced the number of EL-4 leukemia tumor cells (*p* < 0,05) and increased apoptotic cell death (25 mg/kg, *p* < 0.05) in vivo [51].

In another study, CBD (25 µM) likely triggered apoptosis in Jurkat cells through mitochondrial dysfunction, raising ROS levels [52]. Later, the same group demonstrated that tamoxifen sensitizes leukemia cells (T-ALL) to CBD (30 µM), promoting synergism and leading to a significant increase in apoptosis (*p* < 0.0001), with the release of cytochrome c and mitochondrial dysfunction [53]. CBD was cytotoxic to pancreatic tumor cell lines, with IC_50_ values of (20.3 ± 0.4 μM PANC-1) and (18.6 ± 1.2 μM MiaPaCa-2). For normal cells, CBD exerted cytotoxicity at higher doses than for tumor cell lines (IC_50_ 28.62 ± 0.6 μM, H6c7) and (IC_50_ 30.63 ± 1.1 μM), demonstrating a slight selectivity. Exposure to CBD (25 µM) for 48 h induced apoptosis (73.98% PANC-1 and 75.37% MiaPaCa-2) with the activation of caspase-3 (*p* < 0.01 and *p* < 0.001) [54].

Studies using gastric and prostate cancer models have also demonstrated that CBD promoted apoptosis. In gastric cancer cell lines (AGS 4 µM, MKN45 10 µM), CBD regulated Smac and XIAP expression through mitochondrial dysfunction and ER stress. Additionally, the treatment increased the cleaved PARP, caspase-3 (increase 200%, *p* < 0.001 MNK45), caspase-8, and caspase-9 expressions. The action of CBD was remarkably selective for tumor cells (*p* < 0.001) compared to tumor cells, not exerting cytotoxicity up to 10 uM for the normal HFE-145 cell line. In vivo tests, CBD (20 mg/kg) significantly reduced tumor size (*p* < 0.01, n = 5) and weight (*p* < 0.05, n = 5) when compared to control; however, there was no difference in body weight between groups. In addition, TUNEL-positive cells were significantly increased (*p* < 0.001, n = 5) in vivo [55]. In prostate cancer cell lines, such as LNCaP cells and colon cancer cells SW480, CBD (15 μM) induced apoptosis, increasing the levels of cleaved PARP and caspase-3 mediated by CB1 and CB2 receptors [56]. De Petrocellis and collaborators further demonstrated that CBD (10 μM) could activate apoptosis via caspases 3 and 7, independent of CB receptors, in prostate cancer cell lines. CBD (10 mg/kg) combined with other cannabinoids potentiated the effect of docetaxel and bicalutamide, reducing the size of LNCaP xenograft tumors (*p* < 0.05, n = 10) and prolonging the survival rate when combined with bicalutamide [57].plants-14-00585-t003_Table 3Table 3CBD inducing apoptosis in cancer cells.Cancer TypeCell LineDose CurveCBD ConcentrationReceptorMechanismReferencesBladder cancerT241–100 µM30 µMTRPV2↑ Annexin V[49]10–50 µM12 µM-↑ Bax, Cytochrome C, caspase-7↓ PI3K, Akt, mTOR, Bcl-2[50]Breast cancerT-47D,MDA-MB-2311–7 μM3, 5 μM-↑ PPARγ↓ mTOR, cyclin D1[32]MDA-MB-231CBD curve:0.15–10 μM3.72 μM alone and 1 μM in Extracellular Vesicles (EV)-↑ Bax↓ Bcl-2G1 phase cell cycle[34]MDA-MB-231HeLa,SiHa,ME-180DOX curve:0.39–50 μM1 μM in EV combined DOX (500 ηM)-↑ Bax, Caspase-9↓ Bcl-2, mTOR[34,45]50–150 µg/mL3.2, 1.5 µg/mL-↑ p53, Caspase-3, Caspase-7, BaxCervical cancerHCT116,DLD-1,HT-292–8 µMNR6–30 µM-↑ ROS, Noxa, Caspase-3, Caspase-8 and Caspase-9[48]Cólon cancerProstate cancerSW480,LNCaP5–15 µM15 µMCB1 and CB2↑ PARP, Caspase-3[56]Endometrial cancerIshikawa,Hec50co0.01–25 µM5 µMTRPV1↑ ROS, Caspase-3, Caspase-7, c-PARP[44]KSHV-infected HMVECs0.1–100 µM0.1–102.08 and 10 µM-↓ vGPCR, GRO-α, VEGFR-3, VEGF-C[47]Gastric cancer AGS, MKN45, SNU638,NCI-N871–10 µM4 µM and 10 µM-↑ Smac, XIAP, ER stress, cleaved PARP, Caspase-3, Caspase-8, and Caspase-9[55]Glioblastoma U87MG5–20 µM20 μM alone and combined with gamma irradiation-↑ JNK1/2, MAPKP38[35]U87MG, U118MG, T98G5–40 μM, 5–10 Gγ, and 1–5 μM10 μM combined with gamma irradiation and KU60019-↑ TNF/TNFR1, TRAIL/TRAIL-R2, DR5[36]U251, SF126NR0.4 µm combined with THC (1.7 μM)-↑ c-PARP, Caspase-3, Caspase-7, Caspase-9[39]U87MG,U37310–50 µM10 and 25 µM-↑ ROS, Caspase-3, Caspase-8, Caspase-9↓ Glutathione[37,38]GliomaGlioma stem cellsNR4.17 µM combined with THC (0.83 µM) and TMZ (100 µM)-↑ Caspase-3, c-PARP, ROS RIPK3↓ RIPK3, ATP [41]J3TBG, U87MG,U373MG0–20 µg/mL−5.77, 8.2 and 4.94 µg/mLVDAC1
[31]Glioma stem cells (GSCs)0.5–50 µM10 μM CBD + 200 μM carmustine-
-
[10]38323870.1–10 µM2 µM-↑ ROS, p38MAPK↓ Sox2, Id1, p-STAT3[40]Head and neck cancer SCC2510–300 µM30 µM-↑ Annexin V[46]LeukemiaJurkat (T-ALL)5–200 µM25 µM-↑ Mitochondrial dysfunction[52] Jurkat,CCFR-CEM0–100 µMCBD 5 µM + 5 µM Tamoxifen -↑ Mitochondrial Ca^2+^↓ ΔΨm[53]EL-4, Jurkat, MOLT-41.25–105 μMCB2↑ ROS, Caspase-8, Caspase-9, Caspase-3[51]Lung cancerA549H460NR3.47, 2.80 μM-↑ PPARγ, COX2[42] A549,H1299,SCLC H691.5–48 µM10 µM-↑ ROS, Caspase-7, Caspase-3[43] NeuroblastomaSK-N-SH, NUB-62.5–50 µg/mL7.7.55, 100 µg/mL-↑ Caspase-3[7]Pancreatic cancerPANC-1 MiaPaCa-21.52–100 µM25 µM-↑ Caspase-3[54]Prostate cancerLNCaP, DU-145 cellsNR10 µMCB and TRPV independent↑ Caspase-3 and Caspase-7[57]NR, not reported.

#### 2.3.2. Autophagy

In cancer, autophagy typically mitigates cellular stress, maintaining homeostasis and promoting cell survival. However, autophagy can also facilitate the elimination of abnormal cells, including tumor cells. Studies have shown that CBD can activate proteins crucial for autophagy-mediated cell death (Table 4). Research has demonstrated that CBD induces autophagy in glioma, multiple myeloma, and colorectal cancer cell lines. In glioma cells (U87 MG, LN18, U118MG, A172, and U251), CBD triggered autophagic cell death through mitochondrial dysfunction.

In glioblastoma cell lines, CBD (30 µM) promoted a lethal mitophagy-mediated arrest by activating TRPV4, which triggered a Ca2+ influx, a fundamental signal for initiating mitophagy. ER stress and the ATF4-DDIT3-TRIB3-AKT-MTOR axis downstream of TRPV4 were also involved. Furthermore, the authors, using an orthotopic model of glioma, treated the mice with control (n = 6), CBD (15 mg/kg/once per day, n = 8), TMZ (25 mg/kg/once per day, n = 8), and CBD + TMZ (n = 8) for 21 days. It was observed that the combination group presented a significantly slower growth rate and longer survival compared to either agent alone [6].

TRPV2 receptors mediated CBD effects in glioma stem cell lines isolated from adult surgical specimens. The treatment reduced cell viability (GSC #1 IC50 19.4l µM; GSC #30 14.6 µM; GSC #83 19.3l µM at 24 h), and autophagy was evidenced (CBD 10 µM) by the upregulation of cleaved LC3-II and beclin-1 [10].

The same authors later tested CBD (12.5 µM), both alone and in combination with THC, on myeloma cell lines (U266 and RPMI8226). They reported that CBD-THC treatment also activated autophagy, as demonstrated by a rise in the expression of the cleaved LC3-II/LC3-I ratio (*p* < 0.05 vs. control) and a reduction in p62 protein levels (*p* < 0.05 vs. control); this was confirmed using bafilomycin A1 (an autophagy inhibitor), which reversed the cytotoxic effects [58].

In colorectal cancer cell lines resistant to the chemotherapeutic drug oxaliplatin (DLD-1R and colo205R), CBD (4 and 10 µM) overcame drug resistance by activating the autophagic pathway. The treatment promoted mitochondrial dysfunction; reduced phosphorylation of AKT, TOR, AMPK, and NOS3; and enhanced ROS production and LC3 expression. The authors also used a nude BALB/c mice model injected with colo205 R cells. The analyzed groups were oxaliplatin (5 mg/kg, n = 5), CBD (10 mg/kg, n = 5), and their combination (n = 5). After 18 days of treatment, the tumor size of the combination group was significantly lower than the control group (*p* < 0.001) and body weight did not show significant differences in any group [59].plants-14-00585-t004_Table 4Table 4CBD inducing autophagy in cancer cells.Cancer TypeCell LineDose CurveCBC ConcentrationReceptorMechanismReferenceColorectal cancerDLD-1R,colo205R1–7 μM4 and 10 μM-↑ ROS and LC3↓ Akt, TOR, AMPK, NOS3[59]GliomaU87 MG,LN18,U118 MG,A172,U2510–100 μM30 μMTRPV4↑ LC3-II and ER stress, ↓ ATF4, DDIT3, TRIB3, Akt, mTOR [6]Glioma stem cells (GSCs)0.5–50 μM10 μMTRPV2↑ LC3-II, beclin[10]MyelomaU266,RPMI82260–50 μM12.5 μM-↑ LC3-II/LC3-I↓ p62[58]

#### 2.3.3. Necrosis

Unlike apoptosis, necrosis is an uncontrolled form of cell death that occurs in response to harmful stimuli, such as ROS and the dysregulation of cell survival pathways. Two studies included in this review demonstrated that CBD could induce necrosis in multiple myeloma and colorectal adenocarcinoma cell lines (Table 5).

The effect of CBD was enhanced in RPMI8226 and U266 cells transfected with TRPV2, likely due to its interaction with this receptor. When combined with the chemotherapeutic drug bortezomib (3 ng/mL), CBD exerted a synergistic effect, leading to cell cycle arrest and inducing mitochondria-mediated necrosis through ROS generation (approximately 300%). This effect occurred independently of CB1 and CB2 receptors, TRPs, and PPARγ, while regulating ERK, AKT, and both canonical and alternative NF-κB pathways [60].

Cerretani and collaborators evaluated the action of CBD on human colorectal adenocarcinoma cells HT-29 (IC50: 30.0 ± 3.02 µM). The treatment activated cell death by necrosis due to oxidative stress and independent of CB receptors. It increased levels of malondialdehyde (2.6 ± 0.18 nmol/mL) compared to the control (1.6 ± 0.27 nmol/mL) and increased ROS in 62.5% [61].plants-14-00585-t005_Table 5Table 5CBD inducing necrosis in cancer cells.Cancer TypeCell LineDose CurveCBD ConcentrationReceptorMechanismReferenceColorectal cancerHT-29NR30 µM Independent CB receptors↑ ROS and malondialdehyde[61]MyelomaU266,RPMI8226NR20 µM combined with bortezomib (3 ng/mL)TRPV↑ ROS↓ NF-Kβ[60]NR, not reported.

#### 2.3.4. Simultaneous Induction of Apoptosis and Autophagy

Apoptosis and autophagy are homeostatic processes that can lead to cell death through interactions between central proteins in both pathways, which are activated by a single stressor [62]. Studies have shown that CBD can simultaneously induce autophagy and apoptosis in breast, glioblastoma, endometrial, and head and neck cancer cells and leukemia (Table 6).

In breast cancer cell lines, CBD (5 μM) initiated autophagy, which subsequently led to apoptosis as demonstrated by the morphological changes. In MDA-MB231 cells, this dual activation was associated with an increase in LC3-II and cleaved PARP expressions, and ER stress, increasing ROS by a fold change greater than 2 (*p* ≤ 0.01 vs. control). Additionally, CBD inhibited KT/mTOR/4EBP1 and cyclin D signaling pathways. The mitochondrial membrane potential was disrupted, leading to the release of cytochrome c, which triggered both the intrinsic apoptotic pathway and autophagy. The action of CBD on normal human mammary epithelial cells (MCF-10A) was evaluated, and it was observed that although CBD exerted concentration-dependent cytotoxicity, it presented a significant selectivity [63].

In MCF-7 aro cells (estrogen receptor-positive and aromatase-expressing), the high anti-aromatase activity induced by CBD (5 and 10 μM) upregulated beta estrogen receptors and promoted the formation of acidic vesicles, a hallmark of autophagy, as indicated by an increase in the LC3-II/LC3-I ratio (*p* < 0.001). Inhibiting autophagy led to decreased CBD activity and no activation of caspase-9 and caspase-7, suggesting that autophagy is essential for triggering apoptosis and thus promoting more efficient cell death [64].

Torres et al. (2011) demonstrated that a combination of CBD and THC (0.9 µM for each) stimulated autophagy and apoptosis in U87-MG cells. Control cells presented a percentage of LC3 of 2.1 ± 1.2% while CBD + THC treated cells presented 49.1 ± 5.1% (*p* < 0.01 vs. control). Cleaved caspase-3 was also upregulated, with a fold change greater than 15 (*p* < 0.01 vs. control). Furthermore, the combination of THC (0.9 µM), TMZ (75 µM), and CBD (0.9 µM) also enhances autophagy and apoptosis in glioma cells (U87MG and T98G). The effects of the combination of CBD (3.7 mg/kg) with THC (3.7 mg/kg) and TMZ (5 mg/kg) were evaluated on U87MG cell xenografts. The same in vitro findings were observed in vivo, along with a strong reduction in glioma tumor volume and weight (*p* < 0.01 vs. control), inducing cell death by apoptosis (*p* < 0.01 vs. control) and autophagy (*p* < 0.01 vs. control) [65].

Treatment with CBD (10 µM) induced apoptosis via ER stress and autophagy in cells derived from biopsies of primary and recurrent human GBM (GBM10, GBM20, and GBM29. CBD (15 mg/Kg) was also able to increase the survival (*p* = 0.0244) of mice in an orthotopic glioma model [66].

In type I endometrial cancer cell lines (MFE-280 and HEC-1a) and a primary cell line (PCEM002), treatment with CBD (6.33, 23.38, and 6.58 µg/mL, respectively) led to apoptosis, as evidenced by propidium iodide staining and increased annexin V. In contrast, in mixed type I and II cell lines expressing TRPV2 (PCEM004a and PCEM004b), CBD treatment (7.85 and 3.92 µg/mL, respectively) increased LC3-II/LC3-I ratio, particularly in PCEM004a cells, presenting a fold change of approximately 4 after 72 h of treatment (*p* < 0.001), where acidic vesicular organelles indicative of autophagy were observed [67].

CBD (6 µM) also induced apoptosis and autophagy in head and neck cancer cell lines (FaDu, Hep2, SNU-899, and SCC15). DNA damage and apoptosis may have resulted from the upregulation of MCM2, PARP1, or BRCA1. The activation of KLF6 promoted the expression of CDKN1A (p21), which inhibited cyclin D and stimulated the expression of GADD45A/B, leading to cell cycle arrest. Furthermore, the upregulation of autophagic proteins, beclin, and LC3-II was observed. CBD also potentiated the cytotoxic effects of the chemotherapeutic agents cisplatin, 5-fluorouracil, and paclitaxel. Additionally, treatment with CBD (5 mg/kg, n = 4) in the FaDu cell xenograft model reduced the percentage of the tumor area after 21 days (*p* < 0.01 vs. control group), decreased tumor volume (*p* < 0.01 vs. control group), and reduced tumor weight (*p* < 0.05 vs. control group) after 26 days [1].

In acute lymphoblastic leukemia cells (Jurkat, MOLT-3, and CEM), CBD (30 µM) promoted a possible interaction with the VDAC1 receptor, mitochondrial alteration with pore formation, cytochrome C release, and cell death by apoptosis and autophagy. In contrast, CBD did not demonstrate a cytotoxic effect against the normal murine bone marrow cell line (OP9) [68].plants-14-00585-t006_Table 6Table 6CBD inducing both apoptosis and autophagy in cancer cells.Cancer TypeCell LineDose CurveCBD ConcentrationReceptorMechanismReferenceBreast cancerMDA-MB2310–10 μM5 μM-↑ LC3-II, c-PARP, ER stress↓ Akt, mTOR, 4EBP1, Cyclin D(Beclin + Vps34 interaction; Beclin + Bcl2 inhibition)[63]MCF-7aro1–20 μM5 and 10 μM-↑ LC3-II/LC3-I, Caspase-9, Caspase-7[64]Endometrial cancerMFE-280, HEC-1a, and PCEM0020.49–15.72 µg/mL6.33. 23.38, and 6.58 µg/mL-↑ Annexin[67]PCEM004a and PCEM004b
7.85 and 3.92 µg/mL-↑ LC3-II/LC3-IGlioblastomaU87MG and T98GNR0.9 µM CBD combined with 0.9 µM THC-↑ LC3-II, cleaved Caspase-3[65]0.9 µM CBD combined with 0.9 µM THC and 75 µM TMZ-GBM10, GBM20, and GBM291–10 μM10 μM-↑ ER stress[66]Head and neck cancerFaDu, Hep2, SNU-899, and SCC150.1–15 μM6 μM-↑ MCM2, PARP1, BRCA1, KLF6, p21, GADD45A/B↓ Cyclin D[1]LeukemiaJurkat, MOLT-3 and CEM1–100 μM30 μMVDAC1↑ Caspase-3, Caspase-9, Citocromo C, LC3-II[68]NR, not reported.

### 2.4. CBD-Based Drug by the U.S. FDA Trend

The therapeutic value of cannabis, including the cannabinoids CBD and THC, has gained increasing recognition in recent years, with the introduction of new drugs into the market. The Food and Drug Administration (FDA) has approved two synthetic formulations of THC, dronabinol (Marinol^®^, AbbVie Inc., North Chicago, IL, USA), (Syndros^®^, Renaissance Lakewood LLC, Lakewood, NJ, USA) and nabilone (Cesamet™, Valeant Pharmaceuticals International, 3300 Hyland Avenue Costa Mesa, CA, USA), and a CBD-based drug, (Epidiolex, Greenwich Biosciences, Inc., Carlsbad, CA, USA). Studies have demonstrated the efficacy, safety, and tolerability of these drugs [69].

Drugs containing nabilone and dronabinol are used to treat chemotherapy-induced nausea and vomiting [70]. Additionally, dronabinol has been approved to stimulate appetite in patients with acquired immunodeficiency syndrome (AIDS). Studies also indicate how nabilone and dronabinol possess analgesic properties and may be used to manage chronic pain in patients with multiple sclerosis (MS) and cancer [71]. Epidiolex, which consists of 98% CBD, has been approved for the treatment of two severe forms of childhood epilepsy (Dravet syndrome and Lennox–Gastaut syndrome) due to its ability to reduce seizures and prevent motor and cognitive impairment.

In addition to these medications, Sativex, a mouth spray composed of THC and CBD, has been approved in countries such as the United Kingdom, Germany, and Switzerland for relieving muscle stiffness associated with multiple sclerosis (MS) [72]. Furthermore, preclinical studies have explored the efficacy of CBD in treating autoimmune diseases and psychiatric disorders.

Although studies have demonstrated the safety of CBD and THC when used in certain medications, it is important to note that chronic consumption of high doses of cannabis can lead to cannabis hyperemesis syndrome, a condition characterized by persistent nausea and vomiting [73]. Therefore, treatment with cannabis-based medications should be conducted under medical supervision and prescription.

### 2.5. Challenges

Therapeutic interest in CBD has increased in recent years based on positive reports from preclinical and clinical studies. Although it has applications in many therapeutic areas, some limitations and challenges should be considered and discussed.

Clinical studies suggest that CBD is well tolerated and is rarely associated with serious adverse effects, including diarrhea, drowsiness, sedation, and respiratory disorders [74]. However, bioavailability and its therapeutic window must be carefully evaluated. The bioavailability of CBD has been shown to be low among the most conventional routes, such as oral, sublingual, topical, intramuscular, and intravenous. Thus, studies suggest the use of formulations based on lipids, polymers, and nanoparticles to promote improved solubility and delivery of the largest amount administered to the systemic circulation [75]. Although bioavailability is low, the most common form of CBD administration is oral, available in a solution with a concentration of 100 mg/mL. Evidence suggests that the dosage should be applied or suspended according to the patient, such as in cases of liver failure and pregnant women. CBD can cause hepatotoxicity in high daily doses of 300 mg or more and can also be harmful to neurodevelopment during pregnancy [76].

Furthermore, CBD may interfere with the metabolism of other medications, modifying their concentrations in the body. For example, medications with CYP3A4 substrates (immunosuppressants, chemotherapeutics, antidepressants, opioids, and others), when co-administered with CBD, may lead to increased adverse effects related to this substrate. As inhibitors of this substrate (ketoconazole and nefazodone), they may increase the bioavailability of CBD, leading to dose-related toxic effects. Therefore, the use of CBD should be carefully evaluated by healthcare professionals to avoid possible toxic effects [77]. The studies reported here showed that cannabidiol did not cause strong cytotoxic effects on normal cells and mainly increased the survival of the tested animals, although the initial data are promising, and cannabidiol has already been approved by the FDA, further preclinical and clinical studies are needed to ensure safety and carefully define the therapeutic window.

## 3. Discussion

In this review, we explored the antitumor mechanisms of CBD, focusing on its specific molecular targets across various clinically relevant cancer types. Notably, we observed among the articles included in this study that CBD was able to interact mainly with TRPVS receptors, as in endometrial cancer (TRPV1) [44], glioma (TRPV2 and TRPV4) [6,10], bladder cancer (TRPV2) [49], and myeloma (TRPV) [60]. Some studies also reported that the action of CBD is related to the interaction with cannabinoid receptors CB1 and CB2 (colon cancer, prostate cancer, and leukemia) [51,56]. However, there are reports that CBD acts on tumor cells independent of CB and TRPV receptors [57,61]. We also observed that CBD can act on VDAC1 channels, altering the mitochondrial potential to lead to cell death [31,68]. From a physicochemical perspective, CBD is a lipophilic cannabinoid with a logP of 6.1 and a strongest acid pKa of 9.13, making it practically insoluble in water (0.0126 mg/mL) [78]. Cancer progression is often associated with the acidification of the tumor microenvironment (TME), where the pH typically decreases from 7.1–7.7 to 6.7–7.1 [79]. This slight but significant pH shift may enhance CBD’s binding to receptors by increasing the non-ionized fraction of the compound, thereby improving its cellular uptake. In contrast to cationic antineoplastics such as doxorubicin, vincristine, and cisplatin, which may exhibit reduced efficacy in acidic environments, CBD’s neutral form under these conditions could facilitate its therapeutic action.

Optimal cancer treatments aim to selectively destroy malignant cells while sparing healthy ones [80]. By elucidating the mechanisms through which CBD induces cell death specifically in cancer cells, its potential can be harnessed to target cancerous tissue more effectively. This selectivity may result in fewer side effects compared to conventional therapies, such as chemotherapy and radiation, which frequently damage both cancerous and healthy cells [81,82].

Many cancers develop resistance to traditional chemotherapy by evading apoptosis [83,84]. Emerging studies suggest that CBD may overcome these resistance mechanisms by activating alternative cell death pathways, such as autophagy and necroptosis, making it a promising candidate for treating drug-resistant cancers.

Each cancer type exhibits distinct pathological characteristics, with responses to treatment varying based on the involved molecular pathways [85], which explains the wide range of treatment doses reported (0.4 to 30 μM). Consequently, treatment strategies are increasingly shifting toward personalized medicine [86]. Understanding how CBD influences different signaling pathways, depending on cancer type, stage, and tumor genetic makeup, could support the development of more individualized treatment plans [87].

As presented in Table 3, Table 4, Table 5 and Table 6, CBD shows potential for combination with chemotherapeutics to enhance efficacy. Elucidating the molecular mechanisms underlying CBD’s effects can facilitate the design of combination therapies that harness the strengths of multiple treatment modalities, ensuring they act synergistically rather than antagonistically.

Although CBD is already approved for certain conditions (e.g., epilepsy convulsions of patients ≥ 2 years of age with Lennox-Gastaut syndrome or Dravet syndrome [88,89], obtaining regulatory approval for cancer treatment remains challenging due to the limited clinical evidence [90,91]. A robust molecular understanding of CBD is essential for navigating the regulatory landscape and establishing a solid scientific foundation for clinical trials. Such knowledge enables regulatory agencies to assess the safety, efficacy, and therapeutic viability of CBD in cancer treatment. Furthermore, comprehensive evaluation of the bioavailability and pharmacokinetics of CBD in cancer patients is critical, as these factors play a pivotal role in optimizing dosing regimens, maximizing therapeutic efficacy, and ensuring safety in oncologic care.

The major limitation of this review was the limited number of articles identifying the receptors involved in CBD’s anticancer activity. This may be attributed to methodological challenges in in vitro cancer research [92], such as the biochemical complexity of the cell death pathways, the potential involvement of multiple pathways simultaneously, and the limited solubility of CBD in cell culture media.

Moreover, we observed a gap in the literature regarding studies that evaluate CBD’s affinity with vanilloid, cannabinoid, and VDAC1 receptors. In this review, we investigated its interactions using molecular docking. CBD demonstrated high affinity for all the receptors analyzed, with Vina scores ranging from −7.5 to −9.0 kcal/mol [93].

Our molecular docking findings suggest that CBD has a higher affinity for CB2 receptors (−9.0 kcal/mol) compared to TRPV receptors (−7.5 to −7.9 kcal/mol) and VDAC1 receptors (−7.6 kcal/mol). Notably, the phytocannabinoids CBD and THC are recognized as strong agonists of CB1 and CB2 receptors; however, their affinity may vary depending on the tissue analyzed and the receptor expression in cells and tissues [94].

CB2 receptors are expressed in the reproductive and immune systems [95], which may explain CBD’s effects on prostate/colon cancers [56,57] and leukemia [51], respectively. TRPV2 receptors have been implicated in CBD’s action against glioma [10], bladder cancer [49], and myeloma [60]. These receptors are expressed in neurons, smooth muscle, and the immune system [96], corresponding to the cell types and tissues present in these tumors. TRPV1 receptors, expressed in epithelial cells, may account for CBD’s antitumor activity in endometrial cancer [97]. Similarly, TRPV4 receptors, which can be expressed in astrocytes [96], may mediate CBD’s interaction in glioma [6].

Thus, further studies using different cell types and tissues are needed to clarify which receptors are most involved in CBD’s action on tumor cells. However, our docking study predicts that cannabinoid receptors exhibit a higher affinity for CBD.

## 4. Methods

We conducted a comprehensive search in the following databases: PubMed, Scopus, Springer, Medline, Lilacs, and Scielo, using the search terms “cannabidiol” AND “cell death” AND “cancer”. The timeframe for the search spanned from January 1984 to February 2022. We included original research articles in English that reported on in vitro cell death mechanisms and signaling pathways activated by CBD in human tumor cell lines. Articles that did not focus on the in vitro mechanisms of human tumor cell death or signaling pathways activated by CBD, as well as preclinical animal studies, human clinical trials, theses, dissertations, and conference reports, were excluded. Duplicate articles were also removed from consideration. We retrieved 459 articles during the first stage of this search. After removing duplicates and analyzing the titles and abstracts, 416 articles were excluded. Therefore, forty-three articles were read in full. All these remaining articles met the eligibility criteria and were included in this research. The studies covered fourteen tumor types, i.e., breast, glioma, endometrial, head and neck, cervical, lung, sarcoma, colorectal, bladder, leukemia, pancreatic, gastric, prostate, prostate, prostate, and myeloma.

### 4.1. Molecular Docking

CB-Dock2 is a web server publicly available at https://cadd.labshare.cn/cb-dock2/ (accessed on 2 December 2024). It performs highly automatic protein–ligand blind docking using simple steps. Water molecules and heteroatoms were removed (automatically by the server) from the receptor structure, and the other parameters were also automatically filled out. Auto Structure-based Blind Docking was performed, and Vina score and contact residues were obtained [98].

### 4.2. Receptors

Receptor data were obtained from the RCSB Protein Data Bank (https://www.rcsb.org, accessed on 2 December 2024), and the Scientific Name of Source Organism “Homo sapiens” was applied. The protein classification was “Membrane protein” and selection was based on Data Bank score: TRPV1 (PDB ID: 8GF8); TRPV2 (PDB ID: 2F37); TRPV4 (PDB ID: 8T1B); CB1 (PDB ID: 8IKG); CB2 (PDB ID: 8GUQ); VDAC1 (PDB ID: 5XDO).

### 4.3. Ligand

The 2D structure of CBD (644019), called ligand, was obtained from PubChem (https://pubchem.ncbi.nlm.nih.gov, accessed on 2 December 2024) in an SDF.file.

## 5. Conclusions

This review underscores the therapeutic potential of CBD as a promising antitumor agent across various cancer types, particularly through its interaction with transient receptor potential cation channels (TRPVs) and endocannabinoid CB receptors. Our findings suggest that CBD primarily activates the apoptosis pathway while also engaging autophagy and necrosis, offering a multifaceted strategy for inducing cancer cell death. Moreover, the potential for combination therapies that integrate CBD with established chemotherapeutics highlights the importance of further research to investigate these synergies and optimize therapeutic regimens. Although regulatory challenges persist, robust scientific evidence demonstrating CBD’s safety and efficacy of CBD will be crucial for advancing its clinical application in oncology.

## Figures and Tables

**Figure 1 plants-14-00585-f001:**
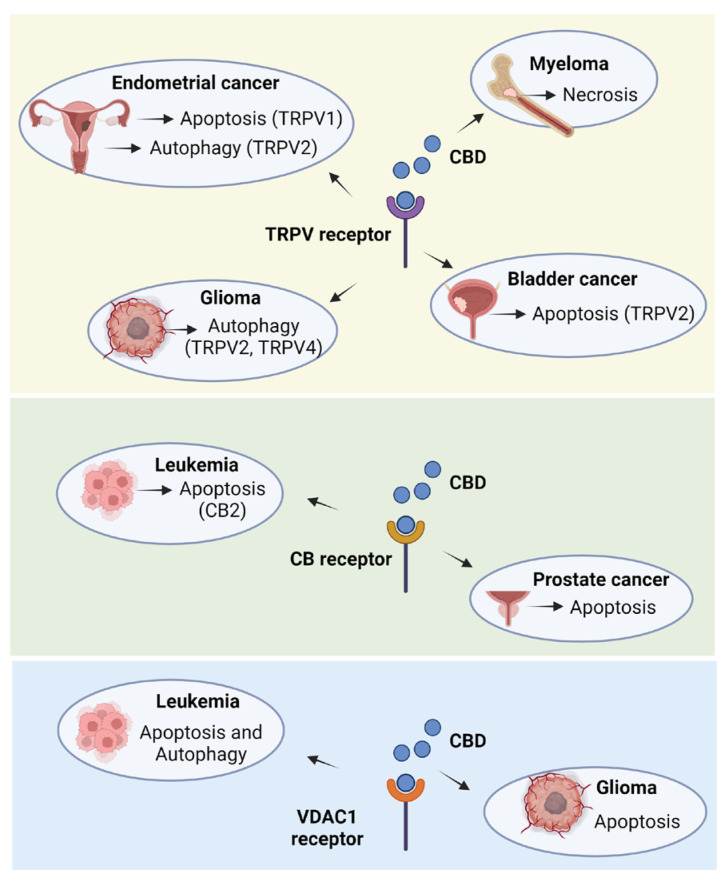
CBD inducing cancer cell death via TRPV, CB and VDAC1 receptors. CBD primarily activated the apoptosis death pathway, and in some cases, it activated autophagy, necrosis, or a combination of apoptosis and autophagy. The receptors responsible for the action of CBD were mostly TRPVs. In endometrial cancer cells, CBD activates apoptosis and autophagy via TRPV1 and TRPV2 receptors, respectively. In glioma, CBD activates autophagy and apoptosis via TRPV2 and TRPV4 receptors. TRPVs may be involved in the activation of necrosis in myeloma and bladder cancer, and TRPV2 is involved in the action of CBD in activating apoptosis. Cannabinoid receptors (CB), although few studies have demonstrated their action in this review, are also involved in the action of CBD in prostate cancer cell lines and leukemia, specifically CB2. Additionally, receptors present in mitochondria (VDAC1) have been shown to be involved in the action of CBD against leukemia and glioma. Created with Biorender.

**Figure 2 plants-14-00585-f002:**
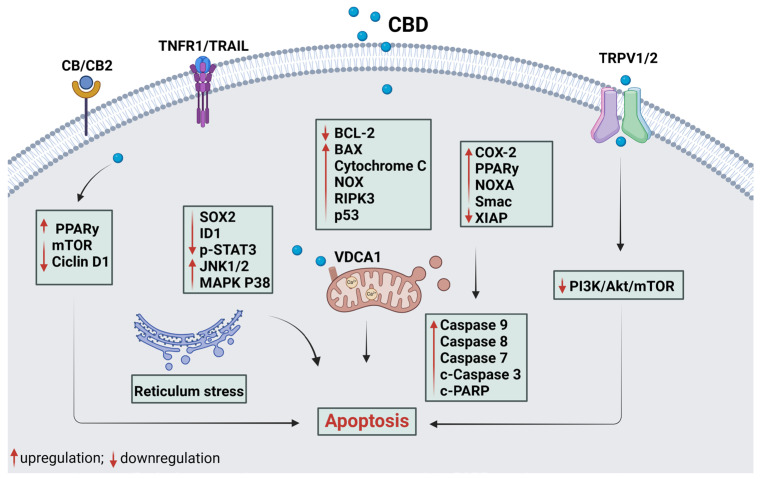
The mechanism of CBD in tumoral cell death by the apoptotic pathway. Cannabidiol binds to TRPV, TNFR1, and CB cannabinoid receptors, specifically CB2, inhibiting cell survival pathways, activating a cascade of cell death proteins, and leading to apoptosis. By activating membrane receptors TRPVs, TNFR1, or CBs, CBD increases PPARy levels and activates apoptosis through cell cycle arrest (inhibition of Cyclin D1 and MTOR). The proteins SOX2, ID1, and p-STAT3 are responsible for increasing cell survival, cell growth, and regulating apoptosis, respectively. Thus, the inhibition of these proteins and the activation of JNK1/2 and MAPK P38 proteins promote cell death and ER stress, leading to apoptosis. CBD also promotes the deregulation of BCL-2 (anti-apoptotic protein) and triggers apoptosis through the activation of BAX, NOX, RIPK3, VDAC1, and p53. CBD alters the permeability of VDAC1, causing Ca^2+^ influx and mitochondrial dysfunction that will promote apoptosis. Thus, CBD promotes the deregulation of the PIK3/Akt/mTOR cell survival pathway, the activation of COX-2, NOXA, and Smac, and the inhibition of XIAP (apoptosis inhibitor protein), leading to intrinsic and/or extrinsic apoptosis with the activation of PARPc and caspases 8, 9, 3, and 7. c-, cleaved; p-, phosphorylated. Red arrows indicate the increase or decrease in protein expression. Black arrows indicate the order of the pathway represented. Created with Biorender.

**Figure 3 plants-14-00585-f003:**
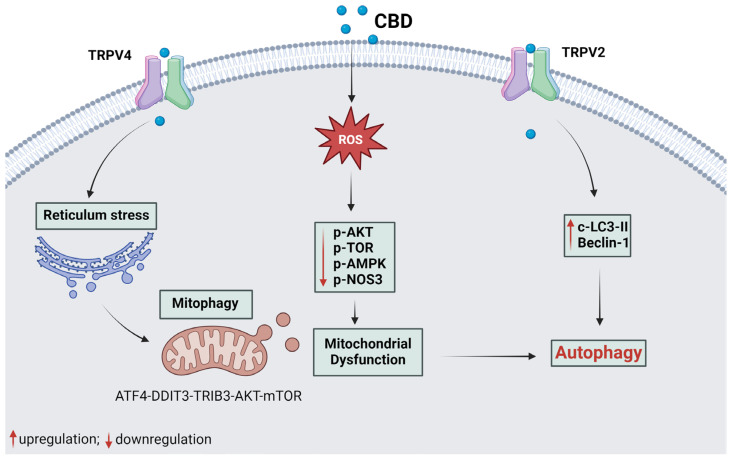
The mechanism of CBD in tumoral cell death by the autophagic pathway. Cannabidiol binds to TRPV2 and TRPV4 receptors and activates proteins responsible for activating autophagy due to oxidative stress of the endoplasmic reticulum and mitochondrial dysfunction. CBD binds to the TRPV4 receptor and leads to cell death by mitophagy through the TRPV4-ATF4-DDIT3-TRIB3-AKT-MTOR axis. Autophagy is activated by CBD due to the presence of ROS, triggering endoplasmic reticulum stress and mitochondrial dysfunction. Thus, CBD promotes the downregulation of cell survival proteins (p-AKT, p-TOR, p-AMPK, and p-NOS3) and upregulates proteins involved in autophagy, such as LC3-II and beclin-1, due to TRPV2 activation. c-, cleaved; p-, phosphorylated. Red arrows indicate the increase or decrease in protein expression. Black arrows indicate the order of the pathway represented. Created with Biorender.

**Figure 4 plants-14-00585-f004:**
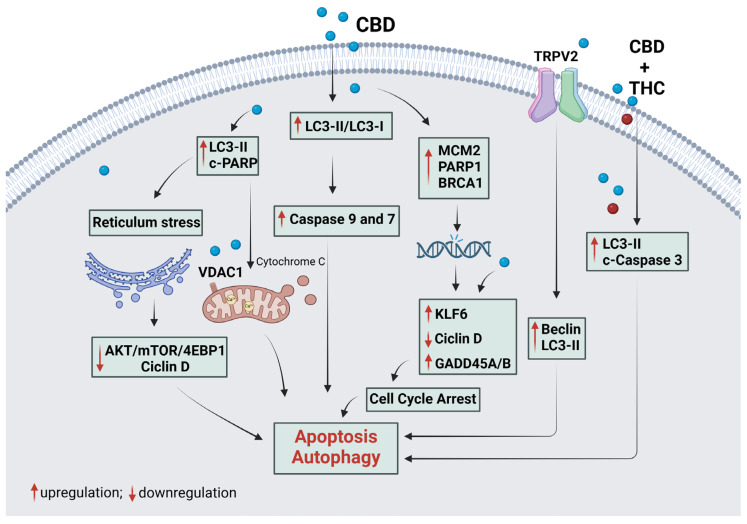
The mechanism of CBD in tumoral cell death via the apoptotic and apoptotic pathways simultaneously. Cannabidiol binds to the TRPV2 receptor, activating apoptosis and autophagy death proteins that work together to promote endoplasmic reticulum stress and DNA damage, leading to cell death. Furthermore, CBD inhibits autophagy, which can act as a survival pathway to promote tumor cell death through apoptosis. CBD can cause DNA damage by upregulating MCM2, PARP1, and BRCA1 and arrests the cell cycle by upregulating KLF6 and GADD454A/B and downregulating Cyclin D. In addition, CBD promotes upregulation of proteins involved in apoptosis (caspase 9, 3, and 7, cytochrome C, MAPK38) and autophagy (LC3-II/I and Beclin) simultaneously. CBD interacts with the VDAC1 receptor, causing mitochondrial alteration with pore formation, cytochrome C release, and cell death by apoptosis and autophagy. Thus, CBD’s upregulation of cell death leads to downregulation of cell survival and growth proteins, such as AKT/mTOR/4EBP1. c-, cleaved; p-, phosphorylated. Red arrows indicate the increase or decrease in protein expression. Black arrows indicate the order of the pathway represented. Created with Biorender.

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
