# Peer review of "Mechanisms of Cell Death Induced by Cannabidiol Against Tumor Cells: A Review of Preclinical Studies"

_plants, 2025, doi:10.3390/plants14040585_

Round 1
Reviewer 1 Report
Comments and Suggestions for Authors
The manuscript “Mechanisms of cell death induced by cannabidiol against tumor cells: a review of in vitro preclinical studies,” written by Ribeiro et al., is a solid work that provides insights into the recent advances of anticancer activity and mechanistic details of cannabidiol.
The authors have collected a full and most recent scope of the antitumor effect of CBD on different types of cell lines, including the probable mechanism of action by providing evidence of receptor-mediated pathways. The work also includes some docking studies indicating strong binding for CBD–receptor interactions.
My main concerns here are the following:
- For all cell studies, the conclusions are difficult to interpret. Some informative parameters (IC50 with standard deviation) should be given to understand the potency of CBD. It is not a quantitative approach to write, “CBD (XXX μM) leads to induced apoptosis.”
- It is also necessary to include the size of the effect, for example, whether ROS increases by 3% or by 300%. Because many compounds that are not effective could lead to insignificant effects on cell proliferation.
- It is another work presenting CBD as the perfect drug. Is it really so? Is there any research on the activity of this compound on healthy cells (HDFa, other fibroblasts)? There are known side effects for Epidiolex. Most probably, it would require higher doses for anticancer activity, and some discussion on the therapeutic window should be included at least briefly.
- The authors should give some additional information that a docking score in the range from –7.5 to –9.0 kcal/mol corresponds to strong binding in terms of CB-Dock2.
I find the submitted paper to be strong and worth publishing in Plants. However, the authors should address the above concerns.
Author Response
- For all cell studies, the conclusions are difficult to interpret. Some informative parameters (IC50 with standard deviation) should be given to understand the potency of CBD. It is not a quantitative approach to write, “CBD (XXX μM) leads to induced apoptosis.”
Response: Thank you very much for your consideration. Some authors consider that IC50 calculations rely on properly fitted dose-response curves using all replicates' data rather than modified individual points (e.g., mean + SD), then, this ensures accuracy and reproducibility in representing the drug's potency.
Nevertheless, for studies that chose to represent IC50 values ​​with standard deviation, we added as requested (Lines: 219-220; 364; 367-368). However, some authors tested fixed doses in cell death assays. For this reason, the values of the tested doses in some articles do not include standard deviation. Regarding the results presented in the tables on protein modulation, we have included only those with statistical significance.
- It is also necessary to include the size of the effect, for example, whether ROS increases by 3% or by 300%. Because many compounds that are not effective could lead to insignificant effects on cell proliferation.
Response: Thank you very much for your consideration. We have written the ROS percentage in all articles that demonstrate this information in the Lines: 210-211; 251; 266; 277; 361; 368; 377.
- It is another work presenting CBD as the perfect drug. Is it really so? Is there any research on the activity of this compound on healthy cells (HDFa, other fibroblasts)? There are known side effects for Epidiolex. Most probably, it would require higher doses for anticancer activity, and some discussion on the therapeutic window should be included at least briefly.
Thank you very much for your contribution. We included the results with CBD in normal cell lines (Lines: 213; 380-382; 423-424). We also wrote a new “Challenges” section (Lines: 467-496) reporting CBD's therapeutic window.
- The authors should give some additional information that a docking score from –7.5 to –9.0 kcal/mol corresponds to strong binding in terms of CB-Dock2.
Response: Thank you very much for your contribution. We have inserted a reference (10.1038/s41598-025-86442-9) indicating the quoted range relative to a strong binding (Line: 545).

Reviewer 2 Report
Comments and Suggestions for Authors
Dear Authors
Congratulations for your work with the title "Mechanisms of cell death induced by cannabidiol against tumor cells: a review of in vitro preclinical studies".
Although you have presented a good piece of work, your data has been limited by the limitations of your approach in line with the title , mainly the subtitle, you have proposed.
Indeed, if you pretend do review in vitro preclinical studies within the framework of cell death induced by cannabidiol against tumor cells, you HAVE to include animal studies. The reason is that by definition for any expert in the area, preclinical studies are research investigations conducted before clinical trials (testing in humans) both in vitro and in vivo.
Additionnally, "preclinical studies" are studies directed to evaluation of the safety, efficacy, and biological activity of a drug, in order to determine whether a potential therapeutic intervention is safe and effective enough to progress to human testing.
Thus, or you could please include adequate preclinical information covered in literature or you could alternatively change the title and the aims, conclusions of your manuscript in order to enable publication after new revision.
Kind regards
Author Response
Congratulations for your work with the title "Mechanisms of cell death induced by cannabidiol against tumor cells: a review of in vitro preclinical studies". Although you have presented a good piece of work, your data has been limited by the limitations of your approach in line with the title, mainly the subtitle, you have proposed.
Comments 1. Indeed, if you pretend do review in vitro preclinical studies within the framework of cell death induced by cannabidiol against tumor cells, you HAVE to include animal studies. The reason is that by definition for any expert in the area, preclinical studies are research investigations conducted before clinical trials (testing in humans) both in vitro and in vivo. Additionnally, "preclinical studies" are studies directed to evaluation of the safety, efficacy, and biological activity of a drug, in order to determine whether a potential therapeutic intervention is safe and effective enough to progress to human testing. Thus, or you could please include adequate preclinical information covered in literature or you could alternatively change the title and the aims, conclusions of your manuscript in order to enable publication after new revision.
Response: Thank you very much for your consideration. We included in vivo studies from the works that were reported in this Review in the Lines: 199-201; 214-218; 224-226; 229-234; 237-240; 268-270; 278-280; 294-297; 324-326; 398-401; 404-405; 418-419.

Reviewer 3 Report
Comments and Suggestions for Authors
The article provides a detailed examination of cannabidiol's (CBD) antitumor mechanisms, but there are areas requiring improvement to enhance its quality, credibility, and impact.
A significant concern involves the figures and images presented in the manuscript. It is not explicitly stated whether these visuals are original creations by the authors or adapted from external sources. The authors need to clarify this and, if the visuals are sourced, provide proper citations and ensure copyright permissions are in place.
The references require meticulous verification to ensure their accuracy and reliability. Also, the overall formatting of the references must adhere to the journal's submission guidelines.
The tables in the article, particularly Tables 1 and 2, include visualizations that may not be necessary for data presentation. The authors are encouraged to remove these visual elements and focus on textual representation of the data.
There are also linguistic and structural issues throughout the manuscript. Some sentences are overly complex or repetitive, making the content less accessible to readers. For example, the discussion on apoptosis and autophagy mechanisms could be rewritten for greater clarity and conciseness. Inconsistent terminology, such as the varying use of "c-Caspase" and "cleaved Caspase," should be standardized to ensure consistency throughout the text.
The methods section of the manuscript lacks sufficient detail about the molecular docking analysis. The authors should expand this section to include information about the software parameters used, receptor preparation steps, and other technical details necessary for reproducibility.
Additionally, the article's limited focus on specific receptors is noted, but the reasons behind the choice of these receptors are not adequately discussed.
Finally, the paper strongly advocates for CBD's therapeutic potential but lacks a balanced discussion of its limitations, potential adverse effects, and comparison with other therapeutic agents.
Author Response
- Reviewer 3)
The article provides a detailed examination of cannabidiol's (CBD) antitumor mechanisms, but there are areas requiring improvement to enhance its quality, credibility, and impact.
- A significant concern involves the figures and images presented in the manuscript. It is not explicitly stated whether these visuals are original creations by the authors or adapted from external sources. The authors need to clarify this and, if the visuals are sourced, provide proper citations and ensure copyright permissions are in place.
Response: Thank you very much for your consideration. All figures presented in the manuscript are original creations of the authors, they were created with Biorender (we have a publication license), we also added the citation “Created with Biorender” in above all the Figures (Line 41, 170, 310, 349, 439).
- The references require meticulous verification to ensure their accuracy and reliability. Also, the overall formatting of the references must adhere to the journal's submission guidelines.
Response: Thank you for your contribution. The references were verified and formatted in the Lines: 667- 922.
- The tables in the article, particularly Tables 1 and 2, include visualizations that may not be necessary for data presentation. The authors are encouraged to remove these visual elements and focus on textual representation of the data.
Response: Thank you for the contribution. We chose to represent the results with visualization of interactions following models published in Plants Journal, below are examples of recent studies that performed molecular docking and added visualization: DOI: 10.3390/plants14020289, 10.3390/plants13233443, 10.3390/plants14010130.
- There are also linguistic and structural issues throughout the manuscript. Some sentences are overly complex or repetitive, making the content less accessible to readers. For example, the discussion on apoptosis and autophagy mechanisms could be rewritten for greater clarity and conciseness.
Response: Thank you very much for your contribution. The entire text has been meticulously checked and the sentences improved (Lines: 22; 130-131), especially in the quoted sections (Lines: 187-296; 321-337; 416-418).
- Inconsistent terminology, such as the varying use of "c-Caspase" and "cleaved Caspase," should be standardized to ensure consistency throughout the text.
Response: Thank you for the contribution. All the text has been checked, and terminology has been standardized and edited in table 6, reference 66.
- The methods section of the manuscript lacks sufficient detail about the molecular docking analysis. The authors should expand this section to include information about the software parameters used, receptor preparation steps, and other technical details necessary for reproducibility.
Response: Thank you for your considerations, we have rewritten the methodology section with the details that were missing (Line 588-602). We also emphasize that CB-Dock2 only requires the receptor file and ligand file, the other parameters were filled in and the molecular docking performed automatically.
- Additionally, the article's limited focus on specific receptors is noted, but the reasons behind the choice of these receptors are not adequately discussed
Response: Thank you for your consideration. We focused on the receptors that were most identified in the studies reported in this Review and were discussed in the Lines: 500-508.
- Finally, the paper strongly advocates for CBD's therapeutic potential but lacks a balanced discussion of its limitations, potential adverse effects, and comparison with other therapeutic agents.
Response: Thank you very much for your contribution. We included the results with CBD in normal cell lines (Lines: 213; 380-382; 423-424). We have also written a new “Challenges” section (Lines: 467-496) reporting CBD's therapeutic window.

Round 2
Reviewer 1 Report
Comments and Suggestions for Authors
Based on the recent amendments, the paper remains of average interest, as older reviews on this topic were clearer and easier to follow. The information in this manuscript is still difficult to comprehend scientifically. I believe the editor should determine whether the current work is suitable for the journal. From my perspective, I would still appreciate it if the authors made the article more scientifically uniform and informative.
For all reported values, it would be necessary to include details in a format such as: "ROS, inhibition properties, protective effect (nature of "protective effect"), etc, in normal cells: XXX ± SD; in cancer cells: XXX ± SD; with the drug: XXX ± SD; with positive/negative control: XXX ± SD. The increase/decrease is XX% (p < 0.05, n = 4)" whenever possible. This should be applied consistently across all properties, not just ROS or any other experiment.
Author Response
(Reviewer 1)
Based on the recent amendments, the paper remains of average interest, as older reviews on this topic were clearer and easier to follow. The information in this manuscript is still difficult to comprehend scientifically. I believe the editor should determine whether the current work is suitable for the journal. From my perspective, I would still appreciate it if the authors made the article more scientifically uniform and informative.
For all reported values, it would be necessary to include details in a format such as: "ROS, inhibition properties, protective effect (nature of "protective effect"), etc, in normal cells: XXX ± SD; in cancer cells: XXX ± SD; with the drug: XXX ± SD; with positive/negative control: XXX ± SD. The increase/decrease is XX% (p < 0.05, n = 4)" whenever possible. This should be applied consistently across all properties, not just ROS or any other experiment.
Response: Thank you for your contribution. We have included more details of the values and results reported by the authors and statistical significance when applicable to make the information more reliable and comparable. For in vivo studies, we were also concerned with including the number of animals.
Lines: 194-196, 198, 201, 204-206, 212-229, 233-236, 240-241, 243-248, 260-263, 266- 276, 279-281, 283-286, 288-289, 292-294, 296-298, 303-304, 306-316, 318-320, 322, 324, 326-332, 336-342, 348, 376-380, 382-383, 386-388, 395-399, 426, 438, 448, 452-455, 459-460, 464, 470-472, and 480-482.
We chose to remove the word "in vitro " be removed from the title, as we now also include in vivo tests.
We also highlight the results with the VDAC1 receptor due to their importance, so we modified Figure 1 (Line: 172-173), Figure 2 (Line: 362-363) and Figure 4 (Line: 500-501), to summarize the effect of CBD mediated by this receptor in leukemia and glioma cells. When mentioned we added it to the Tables (Table 2) and also performed molecular docking to evaluate the interaction with CBD. We inserted the results, methodology and discussion of the new information about the VDAC1 receptor in lines: 107, 116-118, 124-126 (results), 620-623, 625-626 (Discussion) and 667 (Methods).
We have chosen to remove the paper "Inhibition of autophagic flux differently modulates cannabidiol-induced death in 2D and 3D glioblastoma cell cultures" from this Review due to retraction as the editors concluded that "The Editors therefore no longer have confidence in the results and conclusions presented." Therefore, we also modified Figure 4.

Reviewer 3 Report
Comments and Suggestions for Authors
The authors have thoroughly addressed the concerns I raised, providing clear explanations and incorporating the necessary changes.
Author Response
no revisions requested.
Round 3
Reviewer 1 Report
Comments and Suggestions for Authors
I believe the authors did a good job by amending the original text and providing numeric values and statistical data. The paper is good enough to be published.